# Group Sparse Additive Machine

**Hong Chen[1], Xiaoqian Wang[1], Cheng Deng[2], Heng Huang[1]***

[1] Department of Electrical and Computer Engineering, University of Pittsburgh, USA
[2] School of Electronic Engineering, Xidian University, China
chenh@mail.hzau.edu.cn,xqwang1991@gmail.com
chdeng@mail.xidian.edu.cn,heng.huang@pitt.edu

## Abstract

A family of learning algorithms generated from additive models have attracted much attention recently for their flexibility and interpretability in high dimensional data analysis. Among them, learning models with grouped variables have shown competitive performance for prediction and variable selection. However, the previous works mainly focus on the least squares regression problem, not the classification task. Thus, it is desired to design the new additive classification model with variable selection capability for many real-world applications which focus on high-dimensional data classification. To address this challenging problem, in this paper, we investigate the classification with group sparse additive models in reproducing kernel Hilbert spaces. A novel classification method, called as *group sparse additive machine* (GroupSAM), is proposed to explore and utilize the structure information among the input variables. Generalization error bound is derived and proved by integrating the sample error analysis with empirical covering numbers and the hypothesis error estimate with the stepping stone technique. Our new bound shows that GroupSAM can achieve a satisfactory learning rate with polynomial decay. Experimental results on synthetic data and seven benchmark datasets consistently show the effectiveness of our new approach.

## 1 Introduction

The additive models based on statistical learning methods have been playing important roles for the high-dimensional data analysis due to their well performance on prediction tasks and variable selection (deep learning models often don't work well when the number of training data is not large). In essential, additive models inherit the representation flexibility of nonlinear models and the interpretability of linear models. For a learning approach under additive models, there are two key components: the hypothesis function space and the regularizer to address certain restrictions on estimator. Different from traditional learning methods, the hypothesis space used in additive models is relied on the decomposition of input vector. Usually, each input vector $X \in \mathbb{R}^p$ is divided into $p$ parts directly [17, 30, 6, 28] or some subgroups according to prior structural information among input variables [27, 26]. The component function is defined on each decomposed input and the hypothesis function is constructed by the sum of all component functions. Typical examples of hypothesis space include the kernel-based function space [16, 6, 11] and the spline-based function space [13, 15, 10, 30]. Moreover, the Tikhonov regularization scheme has been used extensively for constructing the additive models, where the regularizer is employed to control the complexity of hypothesis space. The examples of regularizer include the kernel-norm regularization associated with the reproducing kernel Hilbert space (RKHS) [5, 6, 11] and various sparse regularization [17, 30, 26].

More recently several group sparse additive models have been proposed to tackle the high-dimensional regression problem due to their nice theoretical properties and empirical effectiveness [15, 10,

---

26]. However, most existing additive model based learning approaches are mainly limited to the least squares regression problem and spline-based hypothesis spaces. Surprisingly, there is no any algorithmic design and theoretical analysis for classification problem with group sparse additive models in RKHS. This paper focuses on filling in this gap on algorithmic design and learning theory for additive models. A novel sparse classification algorithm, called as *group sparse additive machine* (GroupSAM), is proposed under a coefficient-based regularized framework, which is connected to the linear programming support vector machine (LPSVM) [22, 24]. By incorporating the grouped variables with prior structural information and the $\ell_{2,1}$-norm based structured sparse regularizer, the new GroupSAM model can conduct the nonlinear classification and variable selection simultaneously. Similar to the sparse additive machine (SAM) in [30], our GroupSAM model can be efficiently solved via proximal gradient descent algorithm. The main contributions of this paper can summarized in two-fold:

- A new group sparse nonlinear classification algorithm (GroupSAM) is proposed by extending the previous additive regression models to the classification setting, which contains the LPSVM with additive kernel as its special setting. To the best of our knowledge, this is the first algorithmic exploration of additive classification models with group sparsity.

- Theoretical analysis and empirical evaluations on generalization ability are presented to support the effectiveness of GroupSAM. Based on constructive analysis on the hypothesis error, we get the estimate on the excess generalization error, which shows that our GroupSAM model can achieve the fast convergence rate $O(n^{-1})$ under mild conditions. Experimental results demonstrate the competitive performance of GroupSAM over the related methods on both simulated and real data.

Before ending this section, we discuss related works. In [5], support vector machine (SVM) with additive kernels was proposed and its classification consistency was established. Although this method can also be used for grouped variables, it only focuses on the kernel-norm regularizer without addressing the sparseness for variable selection. In [30], the SAM was proposed to deal with the sparse representation on the orthogonal basis of hypothesis space. Despite good computation and generalization performance, SAM does not explore the structure information of input variables and ignores the interactions among variables. More important, different from finite splines approximation in [30], our approach enables us to estimate each component function directly in RKHS. As illustrated in [20, 14], the RKHS-based method is flexible and only depends on few tuning parameters, but the commonly used spline methods need specify the number of basis functions and the sequence of knots.

It should be noticed that the group sparse additive models (GroupSpAM in [26]) also address the sparsity on the grouped variables. However, there are key differences between GroupSAM and GroupSpAM: *1) Hypothesis space.* The component functions in our model are obtained by searching in kernel-based data dependent hypothesis spaces, but the method in [26] uses data independent hypothesis space (not associated with kernel). As shown in [19, 18, 4, 25], the data dependent hypothesis space can provide much more adaptivity and flexibility for nonlinear prediction. The advantage of kernel-based hypothesis space for additive models is also discussed in [14]. *2) Loss function.* The hinge loss used in our classification model is different from the least-squares loss in [26]. *3) Optimization.* Our GroupSAM only needs to construct one component function for each variable group, but the model in [26] needs to find the component functions for each variable in a group. Thus, our method is usually more efficient. Due to the kernel-based component function and non-smooth hinge loss, the optimization of GroupSpAM can not be extended to our model directly. *4) Learning theory.* We establish the generalization bound of GroupSAM by the error estimate technique with data dependent hypothesis spaces, while the error bound is not covered in [26].

Now, we present a brief summary in Table 1 to better illustrate the differences of our GroupSAM with other methods.

The rest of this paper is organized as follows. In next section, we revisit the related classification formulations and propose the new GroupSAM model. Theoretical analysis on generalization error bound is established in Section 3. In Section 4, experimental results on both simulated examples and real data are presented and discussed. Finally, Section 5 concludes this paper.

Table 1: Properties of different additive models.

| | SAM [30] | Group Lasso[27] | GroupSpAM [26] | GroupSAM |
|---|---|---|---|---|
| Hypothesis space | data-independent | data-independent | data-independent | data-dependent |
| Loss function | hinge loss | least-square | least-square | hinge loss |
| Group sparsity | No | Yes | Yes | Yes |
| Generalization bound | Yes | No | No | Yes |

## 2 Group sparse additive machine

In this section, we first revisit the basic background of binary classification and additive models, and then introduce our new GroupSAM model.

Let $\mathcal{Z} := (\mathcal{X}, \mathcal{Y}) \subset \mathbb{R}^{p+1}$, where $\mathcal{X} \subset \mathbb{R}^p$ is a compact input space and $\mathcal{Y} = \{-1, 1\}$ is the set of labels. We assume that the training samples $\mathbf{z} := \{z_i\}_{i=1}^n = \{(x_i, y_i)\}_{i=1}^n$ are independently drawn from an unknown distribution $\rho$ on $\mathcal{Z}$, where each $x_i \in \mathcal{X}$ and $y_i \in \{-1, 1\}$. Let's denote the marginal distribution of $\rho$ on $\mathcal{X}$ as $\rho_{\mathcal{X}}$ and denote its conditional distribution for given $x \in \mathcal{X}$ as $\rho(\cdot|x)$.

For a real-valued function $f : \mathcal{X} \to \mathbb{R}$, we define its induced classifier as $\mathrm{sgn}(f)$, where $\mathrm{sgn}(f)(x) = 1$ if $f(x) \geq 0$ and $\mathrm{sgn}(f)(x) = -1$ if $f(x) < 0$. The prediction performance of $f$ is measured by the misclassification error:

$$\mathcal{R}(f) = \mathrm{Prob}\{Yf(X) \leq 0\} = \int_{\mathcal{X}} \mathrm{Prob}(Y \neq \mathrm{sgn}(f)(x)|x)d\rho_{\mathcal{X}}. \tag{1}$$

It is well known that the minimizer of $\mathcal{R}(f)$ is the Bayes rule:

$$f_c(x) = \mathrm{sgn}\left(\int_{\mathcal{Y}} y d\rho(y|x)\right) = \mathrm{sgn}\left(\mathrm{Prob}(y = 1|x) - \mathrm{Prob}(y = -1|x)\right).$$

Since the Bayes rule involves the unknown distribution $\rho$, it can not be computed directly. In machine learning literature, the classification algorithm usually aims to find a good approximation of $f_c$ by minimizing the empirical misclassification risk:

$$\mathcal{R}_{\mathbf{z}}(f) = \frac{1}{n}\sum_{i=1}^n I(y_i f(x_i) \leq 0), \tag{2}$$

where $I(A) = 1$ if $A$ is true and $0$ otherwise. However, the minimization problem associated with $\mathcal{R}_{\mathbf{z}}(f)$ is NP-hard due to the $0 - 1$ loss $I$. To alleviate the computational difficulty, various convex losses have been introduced to replace the $0 - 1$ loss, *e.g.*, the hinge loss, the least square loss, and the exponential loss [29, 1, 7]. Among them, the hinge loss is the most popular error metric for classification problem due to its nice theoretical properties. In this paper, following [5, 30], we use the hinge loss:

$$\ell(y, f(x)) = (1 - yf(x))_+ = \max\{1 - yf(x), 0\}$$

to measure the misclassification cost. The expected and empirical risks associated with the hinge loss are defined respectively as:

$$\mathcal{E}(f) = \int_{\mathcal{Z}} (1 - yf(x))_+ d\rho(x, y),$$

and

$$\mathcal{E}_{\mathbf{z}}(f) = \frac{1}{n}\sum_{i=1}^n (1 - y_i f(x_i))_+.$$

In theory, the excess misclassification error $\mathcal{R}(\mathrm{sgn}(f)) - \mathcal{R}(f_c)$ can be bounded by the excess convex risk $\mathcal{E}(f) - \mathcal{E}(f_c)$ [29, 1, 7]. Therefore, the classification algorithm usually is constructed under structural risk minimization [22] associated with $\mathcal{E}_{\mathbf{z}}(f)$.

In this paper, we propose a novel group sparse additive machine (GroupSAM) for nonlinear classification. Let $\{1, \cdots, p\}$ be partitioned into $d$ groups. For each $j \in \{1, ..., d\}$, we set $\mathcal{X}^{(j)}$ as the grouped input space and denote $f^{(j)} : \mathcal{X}^{(j)} \to \mathbb{R}$ as the corresponding component function. Usually, the groups can be obtained by prior knowledge [26] or be explored by considering the combinations of input variables [11].

Let each $K^{(j)} : \mathcal{X}^{(j)} \times \mathcal{X}^{(j)} \to \mathbb{R}$ be a Mercer kernel and let $\mathcal{H}_{K^{(j)}}$ be the corresponding RKHS with norm $\| \cdot \|_{K^{(j)}}$. It has been proved in [5] that

$$\mathcal{H} = \Big\{ \sum_{j=1}^{d} f^{(j)} : f^{(j)} \in \mathcal{H}_{K^{(j)}}, 1 \le j \le d \Big\}$$

with norm

$$\|f\|_K^2 = \inf \Big\{ \sum_{j=1}^{d} \|f^{(j)}\|_{K^{(j)}}^2 : f = \sum_{j=1}^{d} f^{(j)} \Big\}$$

is an RKHS associated with the additive kernel $K = \sum_{j=1}^{d} K^{(j)}$.

For any given training set $\mathbf{z} = \{(x_i, y_i)\}_{i=1}^{n}$, the additive model in $\mathcal{H}$ can be formulated as:

$$\bar{f}_{\mathbf{z}} = \operatorname*{arg\,min}_{f = \sum_{j=1}^{d} f^{(j)} \in \mathcal{H}} \Big\{ \mathcal{E}_{\mathbf{z}}(f) + \eta \sum_{j=1}^{d} \tau_j \|f^{(j)}\|_{K^{(j)}}^2 \Big\}, \tag{3}$$

where $\eta = \eta(n)$ is a positive regularization parameter and $\{\tau_j\}$ are positive bounded weights for different variable groups.

The solution $\bar{f}_{\mathbf{z}}$ in (3) has the following representation:

$$\bar{f}_{\mathbf{z}}(x) = \sum_{j=1}^{d} \bar{f}_{\mathbf{z}}^{(j)}(x^{(j)}) = \sum_{j=1}^{d} \sum_{i=1}^{n} \bar{\alpha}_{\mathbf{z},i}^{(j)} y_i K^{(j)}(x_i^{(j)}, x^{(j)}), \ \bar{\alpha}_{\mathbf{z},i}^{(j)} \in \mathbb{R}, \ 1 \le i \le n, \ 1 \le j \le d.$$

Observe that $\bar{f}_{\mathbf{z}}^{(j)}(x) \equiv 0$ is equivalent to $\bar{\alpha}_{\mathbf{z},i}^{(j)} = 0$ for all $i$. Hence, we expect $\|\bar{\alpha}_{\mathbf{z}}^{(j)}\|_2 = 0$ for $\bar{\alpha}_{\mathbf{z}}^{(j)} = (\bar{\alpha}_{\mathbf{z},1}^{(j)}, \cdots, \bar{\alpha}_{\mathbf{z},n}^{(j)})^T \in \mathbb{R}^n$ if the $j$-th variable group is not truly informative. This motivation pushes us to consider the sparsity-induced penalty:

$$\Omega(f) = \inf \Big\{ \sum_{j=1}^{d} \tau_j \|\alpha^{(j)}\|_2 : f = \sum_{j=1}^{d} \sum_{i=1}^{n} \alpha_i^{(j)} y_i K^{(j)}(x_i^{(j)}, \cdot) \Big\}.$$

This group sparse penalty aims at the variable selection [27] and was introduced into the additive regression model [26].

Inspired by learning with data dependent hypothesis spaces [19], we introduce the following hypothesis spaces associated with training samples $\mathbf{z}$:

$$\mathcal{H}_{\mathbf{z}} = \Big\{ f = \sum_{j=1}^{d} f^{(j)} : f^{(j)} \in \mathcal{H}_{\mathbf{z}}^{(j)} \Big\}, \tag{4}$$

where

$$\mathcal{H}_{\mathbf{z}}^{(j)} = \Big\{ f^{(j)} = \sum_{i=1}^{n} \alpha_i^{(j)} K^{(j)}(x_i^{(j)}, \cdot) : \alpha_i^{(j)} \in \mathbb{R} \Big\}.$$

Under the group sparse penalty and data dependent hypothesis space, the group sparse additive machine (GroupSAM) can be written as:

$$f_{\mathbf{z}} = \operatorname*{arg\,min}_{f \in \mathcal{H}_{\mathbf{z}}} \Big\{ \frac{1}{n} \sum_{i=1}^{n} (1 - y_i f(x_i))_+ + \lambda \Omega(f) \Big\}, \tag{5}$$

where $\lambda > 0$ is a regularization parameter.

Let's denote $\alpha^{(j)} = (\alpha_1^{(j)}, \cdots, \alpha_n^{(j)})^T$ and $\mathbf{K}_i^{(j)} = (K^{(j)}(x_1^{(j)}, x_i^{(j)}), \cdots, K^{(j)}(x_n^{(j)}, x_i^{(j)}))^T$. The GroupSAM in (5) can be rewritten as:

$$f_\mathbf{z} = \sum_{j=1}^d f_\mathbf{z}^{(j)} = \sum_{j=1}^d \sum_{t=1}^n \alpha_{\mathbf{z},t}^{(j)} K^{(j)}(x_t^{(j)}, \cdot),$$

with

$$\{\alpha_\mathbf{z}^{(j)}\} = \underset{\alpha^{(j)} \in \mathbb{R}^n, 1 \le j \le d}{\arg\min} \left\{ \frac{1}{n} \sum_{i=1}^n \left(1 - y_i \sum_{j=1}^d (\mathbf{K}_i^{(j)})^T \alpha^{(j)}\right)_+ + \lambda \sum_{j=1}^d \tau_j \|\alpha^{(j)}\|_2 \right\}. \tag{6}$$

The formulation (6) transforms the function-based learning problem (5) into a coefficient-based learning problem in a finite dimensional vector space. The solution of (5) is spanned naturally by the kernelized functions $\{K^{(j)}(\cdot, x_i^{(j)}))\}$, rather than B-Spline basis functions [30]. When $d = 1$, our GroupSAM model degenerates to the special case which includes the LPSVM loss and the sparsity regularization term. Compared with LPSVM [22, 24] and SVM with additive kernels [5], our GroupSAM model imposes the sparsity on variable groups to improve the prediction interpretation of additive classification model.

For given $\{\tau_j\}$, the optimization problem of GroupSAM can be computed efficiently via an accelerated proximal gradient descent algorithm developed in [30]. Due to space limitation, we don't recall the optimization algorithm here again.

## 3   Generalization error bound

In this section, we will derive the estimate on the excess misclassification error $\mathcal{R}(\mathrm{sgn}(f_\mathbf{z})) - \mathcal{R}(f_c)$. Before providing the main theoretical result, we introduce some necessary assumptions for learning theory analysis.

**Assumption A.** The intrinsic distribution $\rho$ on $\mathcal{Z} := \mathcal{X} \times \mathcal{Y}$ satisfies the Tsybakov noise condition with exponent $0 \le q \le \infty$. That is to say, for some $q \in [0, \infty)$ and $\Delta > 0$,

$$\rho_\mathcal{X}\Big(\{x \in \mathcal{X} : |\mathrm{Prob}(y=1|x) - \mathrm{Prob}(y=-1|x)| \le \Delta t\}\Big) \le t^q, \forall t > 0. \tag{7}$$

The Tsybakov noise condition was proposed in [21] and has been used extensively for theoretical analysis of classification algorithms [24, 7, 23, 20]. Indeed, (7) holds with exponent $q = 0$ for any distribution and with $q = \infty$ for well separated classes.

Now we introduce the empirical covering numbers [8] to measure the capacity of hypothesis space.

**Definition 1** *Let $\mathcal{F}$ be a set of functions on $\mathcal{Z}$ with $\mathbf{u} = \{u_i\}_{i=1}^k \subset \mathcal{Z}$. Define the $\ell_2$-empirical metric as $\ell_{2,\mathbf{u}}(f,g) = \{\frac{1}{n} \sum_{t=1}^k (f(u_t) - g(u_t))^2\}^{\frac{1}{2}}$. The covering number of $\mathcal{F}$ with $\ell_2$-empirical metric is defined as $\mathcal{N}_2(\mathcal{F}, \varepsilon) = \sup_{n \in \mathbb{N}} \sup_{\mathbf{u} \in \mathcal{X}^n} \mathcal{N}_{2,\mathbf{u}}(\mathcal{F}, \varepsilon)$, where*

$$\mathcal{N}_{2,\mathbf{u}}(\mathcal{F}, \varepsilon) = \inf\left\{l \in \mathbb{N} : \exists \{f_i\}_{i=1}^l \subset \mathcal{F} \ s.t. \ \mathcal{F} = \bigcup_{i=1}^l \{f \in \mathcal{F} : \ell_{2,\mathbf{u}}(f, f_i) \le \varepsilon\}\right\}.$$

Let $\mathcal{B}_r = \{f \in \mathcal{H}_K : \|f\|_K \le r\}$ and $\mathcal{B}_r^{(j)} = \{f^{(j)} \in \mathcal{H}_{K^{(j)}} : \|f^{(j)}\|_{K^{(j)}} \le r\}$.

**Assumption B.** Assume that $\kappa = \sum_{j=1}^d \sup_{x^{(j)}} \sqrt{K^{(j)}(x^{(j)}, x^{(j)})} < \infty$ and for some $s \in (0, 2), c_s > 0$,

$$\log \mathcal{N}_2(\mathcal{B}_1^{(j)}, \varepsilon) \le c_s \varepsilon^{-s}, \ \forall \varepsilon > 0, \ j \in \{1, ..., d\}.$$

It has been asserted in [6] that under Assumption B the following holds:

$$\log \mathcal{N}_2(\mathcal{B}_1, \varepsilon) \le c_s d^{1+s} \varepsilon^{-s}, \ \forall \varepsilon > 0.$$

It is worthy noticing that the empirical covering number has been studied extensively in learning theory literatures [8, 20]. Detailed examples have been provided in Theorem 2 of [19], Lemma 3 of [18], and Examples 1, 2 of [9]. The capacity condition of additive assumption space just depends on the dimension of subspace $\mathcal{X}^{(j)}$. When $K^{(j)} \in C^\nu(\mathcal{X}^{(j)} \times \mathcal{X}^{(j)})$ for every $j \in \{1, \cdots, d\}$, the theoretical analysis in [19] assures that Assumption B holds true for:

$$
s = \begin{cases}
\frac{2d_0}{d_0+2\nu}, & \nu \in (0,1]; \\
\frac{2d_0}{d_0+\nu}, & \nu \in [1, 1+d_0/2]; \\
\frac{d_0}{\nu}, & \nu \in (1+d_0/2, \infty).
\end{cases}
$$

Here $d_0$ denotes the maximum dimension among $\{\mathcal{X}^{(j)}\}$.

With respect to (3), we introduce the data-free regularized function $f_\eta$ defined by:

$$
f_\eta = \underset{f=\sum_{j=1}^d f^{(j)} \in \mathcal{H}}{\arg\min} \left\{ \mathcal{E}(f) + \eta \sum_{j=1}^d \tau_j \|f^{(j)}\|_{K^{(j)}}^2 \right\}. \tag{8}
$$

Inspired by the analysis in [6], we define:

$$
D(\eta) = \mathcal{E}(f_\eta) - \mathcal{E}(f_c) + \eta \sum_{j=1}^d \tau_j \|f_\eta^{(j)}\|_{K^{(j)}}^2 \tag{9}
$$

as the approximation error, which reflects the learning ability of hypothesis space $\mathcal{H}$ under Tikhonov regularization scheme.

The following approximation condition has been studied and used extensively for classification problems, such as [3, 7, 24, 23]. Please see Examples 3 and 4 in [3] for the explicit version for Soblov kernel and Gaussian kernel induced reproducing kernel Hilbert space.

**Assumption C.** There exists an exponent $\beta \in (0,1)$ and a positive constant $c_\beta$ such that:

$$
D(\eta) \le c_\beta \eta^\beta, \forall \eta > 0.
$$

Now we introduce our main theoretical result on the generalization bound as follows.

**Theorem 1** *Let $0 < \min_j \tau_j \le \max_j \tau_j \le c_0 < \infty$ and Assumptions A-C hold true. Take $\lambda = n^{-\theta}$ in (5) for $0 < \theta \le \min\{\frac{2-s}{2s}, \frac{3+5\beta}{2-2\beta}\}$. For any $\delta \in (0,1)$, there exists a constant $C$ independent of $n, \delta$ such that*

$$
\mathcal{R}(\mathrm{sgn}(f_\mathbf{z})) - \mathcal{R}(f_c) \le C \log(3/\delta) n^{-\vartheta}
$$

*with confidence $1-\delta$, where*

$$
\vartheta = \min\left\{ \frac{q+1}{q+2}, \frac{\beta(2\theta+1)}{2\beta+2}, \frac{(q+1)(2-s-2s\theta)}{4+2q+sq}, \frac{3+5\beta+2\beta\theta-2\theta}{4+4\beta} \right\}.
$$

Theorem 1 demonstrates that GroupSAM in (5) can achieve the convergence rate with polynomial decay under mild conditions in hypothesis function space. When $q \to \infty$, $\beta \to 1$, and each $K^{(j)} \in C^\infty$, the error decay rate of GroupSAM can arbitrarily close to $O(n^{-\min\{1, \frac{1+2\theta}{4}\}})$. Hence, the fast convergence rate $O(n^{-1})$ can be obtained under proper selections on parameters. To verify the optimal bound, we need provide the lower bound for the excess misclassification error. This is beyond the main focus of this paper and we leave it for future study.

Additionally, the consistency of GroupSAM can be guaranteed with the increasing number of training samples.

**Corollary 1** *Under conditions in Theorem 1, there holds $\mathcal{R}(\mathrm{sgn}(f_\mathbf{z})) - \mathcal{R}(f_c) \to 0$ as $n \to \infty$.*

To better understand our theoretical result, we compare it with the related works as below:

*1) Compared with group sparse additive models*. Although the asymptotic theory of group sparse additive models has been well studied in [15, 10, 26], all of them only consider the regression task under the mean square error criterion and basis function expansion. Due to the kernel-based component function and non-smooth hinge loss, the previous analysis cannot be extended to GroupSAM directly.

*2) Compared with classification with additive models*. In [30], the convergence rate is presented for sparse additive machine (SAM), where the input space $\mathcal{X}$ is divided into $p$ subspaces directly without considering the interactions among variables. Different to the sparsity on variable groups in this paper, SAM is based on the sparse representation of orthonormal basis similar with [15]. In [5], the consistency of SVM with additive kernel is established, where the kernel-norm regularizer is used. However, the sparsity on variables and the learning rate are not investigated in previous articles.

*3) Compared with the related analysis techniques*. While the analysis technique used here is inspired from [24, 23], it is the first exploration for additive classification model with group sparsity. In particular, the hypothesis error analysis develops the stepping stone technique from the $\ell_1$-norm regularizer to the group sparse $\ell_{2,1}$-norm regularizer. Our analysis technique also can be applied to other additive models. For example, we can extend the shrunk additive regression model in [11] to the sparse classification setting and investigate its generalization bound by the current technique.

**Proof sketches of Theorem 1**

To get tight error estimate, we introduce the clipping operator $\pi(f)(x) = \max\{-1, \min\{f(x), 1\}\}$, which has been widely used in learning theory literatures, such as [7, 20, 24, 23]. Since $\mathcal{R}(\mathrm{sgn}(f_{\mathbf{z}})) - \mathcal{R}(f_c)$ can be bounded by $\mathcal{E}(\pi(f_{\mathbf{z}})) - \mathcal{E}(f_c)$, we focus on bounding the excess convex risk.

Using $f_\eta$ as the intermediate function, we can obtain the following error decomposition.

**Proposition 1** *For $f_{\mathbf{z}}$ defined in (5), there holds*

$$\mathcal{R}(\mathrm{sgn}(f_{\mathbf{z}})) - \mathcal{R}(f_c) \leq \mathcal{E}(\pi(f_{\mathbf{z}})) - \mathcal{E}(f_c) \leq E_1 + E_2 + E_3 + D(\eta),$$

*where $D(\eta)$ is defined in (9),*

$$
\begin{aligned}
E_1 &= \mathcal{E}(\pi(f_{\mathbf{z}})) - \mathcal{E}(f_c) - \big(\mathcal{E}_{\mathbf{z}}(\pi(f_{\mathbf{z}})) - \mathcal{E}_{\mathbf{z}}(f_c)\big), \\
E_2 &= \mathcal{E}_{\mathbf{z}}(f_\eta) - \mathcal{E}_{\mathbf{z}}(f_c) - \big(\mathcal{E}_{\mathbf{z}}(f_\eta) - \mathcal{E}(f_c)\big),
\end{aligned}
$$

*and*

$$E_3 = \mathcal{E}_{\mathbf{z}}(\pi(f_{\mathbf{z}})) + \lambda\Omega(f_{\mathbf{z}}) - \Big(\mathcal{E}_{\mathbf{z}}(f_\eta) + \eta\sum_{j=1}^{d}\tau_j\|f_\eta^{(j)}\|_{K^{(j)}}^2\Big).$$

In learning theory literature, $E_1 + E_2$ is called as the sample error and $E_3$ is named as the hypothesis error. Detailed proofs for these error terms are provided in the supplementary materials.

The upper bound of hypothesis error demonstrates that the divergence induced from regularization and hypothesis space tends to zero as $n \to \infty$ under proper selected parameters. To estimate the hypothesis error $E_3$, we choose $\bar{f}_{\mathbf{z}}$ as the stepping stone function to bridge $\mathcal{E}_{\mathbf{z}}(\pi(f_{\mathbf{z}})) + \lambda\Omega(f_{\mathbf{z}})$ and $\mathcal{E}_{\mathbf{z}}(f_\eta) + \lambda\sum_{j=1}^{d}\tau_j\|f_\eta^{(j)}\|_{K^{(j)}}^2$. The proof is inspired from the stepping stone technique for support vector machine classification [24]. Notice that our analysis is associated with the $\ell_{2,1}$-norm regularizer while the previous analysis just focuses on the $\ell_1$-norm regularization.

The error term $E_1$ reflects the divergence between the expected excess risk $\mathcal{E}(\pi(f_{\mathbf{z}})) - \mathcal{E}(f_c)$ and the empirical excess risk $\mathcal{E}_{\mathbf{z}}(\pi(f_{\mathbf{z}})) - \mathcal{E}_{\mathbf{z}}(f_c)$. Since $f_{\mathbf{z}}$ involves any given $\mathbf{z} = \{(x_i, y_i)\}_{i=1}^n$, we introduce the concentration inequality in [23] to bound $E_1$. We also bound the error term $E_2$ in terms of the one-side Bernstein inequality [7].

## 4  Experiments

To evaluate the performance of our proposed GroupSAM model, we compare our model with the following methods: SVM (linear SVM with $\ell_2$-norm regularization), L1SVM (linear SVM with $\ell_1$-norm regularization), GaussianSVM (nonlinear SVM using Gaussian kernel), SAM (Sparse Additive Machine) [30], and GroupSpAM (Group Sparse Additive Models) [26] which is adapted to the classification setting.

Table 2: Classification accuracy comparison on the synthetic data. The upper half shows the results with 24 features groups, while the lower half corresponds to the results with 300 feature groups. The table shows the average classification accuracy and the standard deviation in 2-fold cross validation.

|  | SVM | GaussianSVM | L1SVM | SAM | GroupSpAM | GroupSAM |
|---|---|---|---|---|---|---|
| $\sigma = 0.8$ | 0.943±0.011 | 0.935±0.028 | 0.925±0.035 | 0.895±0.021 | 0.880±0.021 | **0.953±0.018** |
| $\sigma = 0.85$ | 0.943±0.004 | 0.938±0.011 | 0.938±0.004 | 0.783±0.088 | 0.868±0.178 | **0.945±0.000** |
| $\sigma = 0.9$ | 0.935±0.014 | 0.925± 0.007 | 0.938±0.011 | 0.853± 0.117 | 0.883±0.011 | **0.945±0.007** |
| $\sigma = 0.8$ | 0.975±0.035 | 0.975±0.035 | 0.975±0.035 | 0.700±0.071 | 0.275±0.106 | **1.000±0.000** |
| $\sigma = 0.85$ | 0.975±0.035 | 0.975±0.035 | 0.975±0.035 | 0.600±0.141 | 0.953±0.004 | **1.000±0.000** |
| $\sigma = 0.9$ | 0.975±0.035 | 0.975±0.035 | 0.975±0.035 | 0.525±0.035 | 0.983±0.004 | **1.000±0.000** |

As for evaluation metric, we calculate the classification accuracy, *i.e.,* percentage of correctly labeled samples in the prediction. In comparison, we adopt 2-fold cross validation and report the average performance of each method.

We implement SVM, L1SVM and GaussianSVM using the LIBSVM toolbox [2]. We determine the hyper-parameter of all models, *i.e.,* parameter $C$ of SVM, L1SVM and GaussianSVM, parameter $\lambda$ of SAM, parameter $\lambda$ of GroupSpAM, parameter $\lambda$ in Eq. (6) of GroupSAM, in the range of $\{10^{-3}, 10^{-2}, \ldots, 10^{3}\}$. We tune the hyper-parameters via 2-fold cross validation on the training data and report the best parameter *w.r.t.* classification accuracy of each method. In the accelerated proximal gradient descent algorithm for both SAM and GroupSAM, we set $\mu = 0.5$, and the number of maximum iterations as 2000.

## 4.1 Performance comparison on synthetic data

We first examine the classification performance on the synthetic data as a sanity check. Our synthetic data is randomly generated as a mixture of Gaussian distributions. In each class, data points are sampled *i.i.d.* from a multivariate Gaussian distribution with the covariance being $\sigma I$, with $I$ as the identity matrix. This setting indicates independent covariates of the data. We set the number of classes to be 4, the number of samples to be 400, and the number of dimensions to be 24. We set the value of $\sigma$ in the range of $\{0.8, 0.85, 0.9\}$ respectively. Following the experimental setup in [31], we make three replicates for each feature in the data to form 24 feature groups (each group has three replicated features). We randomly pick 6 feature groups to generate the data such that we can evaluate the capability of GroupSAM in identifying truly useful feature groups. To make the classification task more challenging, we add random noise drawn from uniform distribution $\mathcal{U}(0, \theta)$ where $\theta$ is 0.8 times the maximum value in the data. In addition, we test on a high-dimensional case by generating 300 feature groups (*e.g.,* a total of 900 features) with 40 samples in a similar approach.

We summarize the classification performance comparison on the synthetic data in Table 2. From the experimental results we notice that GroupSAM outperforms other approaches under all settings. This comparison verifies the validity of our method. We can see that GroupSAM significantly improves the performance of SAM, which shows that the incorporation of group information is indeed beneficial for classification. Moreover, we can notices the superiority of GroupSAM over GroupSpAM, which illustrates that our GroupSAM model is more suitable for classiciation. We also present the comparison of feature groups in Table 3. For illustration purpose, we use the case with 24 feature groups as an example. Table 3 shows that the feature groups identified by GroupSAM are exactly the same as the ground truth feature groups used for synthetic data generation. Such results further demonstrate the effectiveness of GroupSAM method, from which we know GroupSAM is able to select the truly informative feature groups thus improve the classification performance.

## 4.2 Performance comparison on benchmark data

In this subsection, we use 7 benchmark data from UCI repository [12] to compare the classification performance of different methods. The 7 benchmark data includes: Ecoli, Indians Diabetes, Breast Cancer, Stock, Balance Scale, Contraceptive Method Choice (CMC) and Fertility. Similar to the settings in synthetic data, we construct feature groups by replicating each feature for 3 times. In each

Table 3: Comparison between the true feature group ID (used for data generation) and the selected feature group ID by our GroupSAM method on the synthetic data. Order of the true feature group ID does not represent the order of importance.

| | True Feature Group IDs | Selected Feature Group IDs via GroupSAM |
|---|---|---|
| $\sigma = 0.8$ | 2,3,4,8,10,17 | 3,10,17,8,2,4 |
| $\sigma = 0.85$ | 1,5,10,12,17,21 | 5,12,17,21,1,10 |
| $\sigma = 0.9$ | 2,6,7,9,12,22 | 6,22,7,9,2,12 |

Table 4: Classification accuracy comparison on the benchmark data. The table shows the average classification accuracy and the standard deviation in 2-fold cross validation.

| | SVM | GaussianSVM | L1SVM | SAM | GroupSpAM | GroupSAM |
|---|---|---|---|---|---|---|
| Ecoli | 0.815±0.054 | 0.818±0.049 | 0.711±0.051 | 0.816±0.039 | 0.771±0.009 | **0.839±0.028** |
| Indians Diabetes | 0.651±0.000 | 0.652±0.002 | 0.638±0.018 | 0.652±0.000 | 0.643±0.004 | **0.660±0.013** |
| Breast Cancer | **0.968±0.017** | 0.965±0.017 | 0.833±0.008 | 0.833±0.224 | 0.958±0.027 | 0.966±0.014 |
| Stock | 0.913±0.001 | 0.911±0.002 | 0.873±0.001 | 0.617±0.005 | 0.875±0.005 | **0.917±0.005** |
| Balance Scale | 0.864± 0.003 | 0.869±0.004 | 0.870±0.003 | 0.763±0.194 | 0.848±0.003 | **0.893±0.003** |
| CMC | 0.420± 0.011 | 0.445±0.015 | 0.437±0.014 | 0.427±0.000 | 0.433±0.003 | **0.456±0.003** |
| Fertility | **0.880± 0.000** | **0.880±0.000** | 0.750±0.184 | 0.860±0.028 | 0.780±0.000 | **0.880±0.000** |

feature group, we add random noise drawn from uniform distribution $\mathcal{U}(0, \theta)$ where $\theta$ is 0.3 times the maximum value in each data.

We display the comparison results in Table 4. We find that GroupSAM performs equal or better than the compared methods in all benchmark datasets. Compared with SVM and L1SVM, our method uses additive model to incorporate nonlinearity thus is more appropriate to find the complex decision boundary. Moreover, the comparison with Gaussian SVM and SAM illustrates that by involving the group information in classification, GroupSAM makes better use of the structure information among features such that the classification ability can be enhanced. Compared with GroupSpAM, our GroupSAM model is proposed in data dependent hypothesis spaces and employs hinge loss in the objective, thus is more suitable for classification.

## 5    Conclusion

In this paper, we proposed a novel group sparse additive machine (GroupSAM) by incorporating the group sparsity into the additive classification model in reproducing kernel Hilbert space. By developing the error analysis technique with data dependent hypothesis space, we obtain the generalization error bound of the proposed GroupSAM, which demonstrates our model can achieve satisfactory learning rate under mild conditions. Experimental results on both synthetic and real-world benchmark datasets validate the algorithmic effectiveness and support our learning theory analysis. In the future, it is interesting to investigate the learning performance of robust group sparse additive machines with loss functions induced by quantile regression [6, 14].

## Acknowledgments

This work was partially supported by U.S. NSF-IIS 1302675, NSF-IIS 1344152, NSF-DBI 1356628, NSF-IIS 1619308, NSF-IIS 1633753, NIH AG049371. Hong Chen was partially supported by National Natural Science Foundation of China (NSFC) 11671161. We are grateful to the anonymous NIPS reviewers for the insightful comments.

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
