[Supplementary Material]

# Supplementary Material to "Group Sparse Additive Machine"

## 1 Technical proof of Theorem 1

For feasibility, we recall the error decomposition in Section 3 as below.

**Proposition 1** *For $f_{\mathbf{z}}$ defined in Section 2, there holds*

$$
\begin{aligned}
\mathcal{R}(\mathrm{sgn}(f_{\mathbf{z}})) - \mathcal{R}(f_c) &\leq \mathcal{E}(\pi(f_{\mathbf{z}})) - \mathcal{E}(f_c) \\
&\leq E_1 + E_2 + E_3 + D(\eta),
\end{aligned}
$$

*where $D(\eta)$ is defined in Section 3,*

$$
\begin{aligned}
E_1 &= \mathcal{E}(\pi(f_{\mathbf{z}})) - \mathcal{E}(f_c) - \big(\mathcal{E}_{\mathbf{z}}(\pi(f_{\mathbf{z}})) - \mathcal{E}_{\mathbf{z}}(f_c)\big), \quad(1) \\
E_2 &= \mathcal{E}_{\mathbf{z}}(f_\eta) - \mathcal{E}_{\mathbf{z}}(f_c) - \big(\mathcal{E}_{\mathbf{z}}(f_\eta) - \mathcal{E}(f_c)\big), \quad(2)
\end{aligned}
$$

*and*

$$
E_3 = \mathcal{E}_{\mathbf{z}}(\pi(f_{\mathbf{z}})) + \lambda\Omega(f_{\mathbf{z}}) - \big(\mathcal{E}_{\mathbf{z}}(f_\eta) + \eta\sum_{j=1}^{d}\tau_j\|f_\eta^{(j)}\|_{K^{(j)}}^2\big). \quad(3)
$$

### 1.1 Hypothesis error estimate

To estimate the hypothesis error $E_3$, we choose $\bar{f}_{\mathbf{z}}$ in Section 2 as the stepping stone function to bridge $\mathcal{E}_{\mathbf{z}}(\pi(f_{\mathbf{z}})) + \lambda\Omega(f_{\mathbf{z}})$ and $\mathcal{E}_{\mathbf{z}}(f_\eta) + \lambda\sum_{j=1}^{d}\tau_j\|f_\eta^{(j)}\|_{K^{(j)}}^2$. The proof is inspired from the stepping stone technique for support vector machine classification [6, 2].

The optimization framework for $\bar{f}_{\mathbf{z}}$ can be rewritten as the following quadratic programming optimization problem:

$$
\begin{aligned}
\min_{f^{(j)}} \quad & \frac{1}{n}\sum_{i=1}^{n}\xi_i + \eta\sum_{j=1}^{d}\tau_j\langle f^{(j)}, f^{(j)}\rangle_{K^{(j)}}, \\
s.t \quad & y_i\sum_{j=1}^{d}\langle f^{(j)}, K^{(j)}(x_i^{(j)},\cdot)\rangle_{K^{(j)}} \geq 1 - \xi_i, \\
& \xi_i \geq 0, i = 1,...,n.
\end{aligned}
$$

We define the Lagrangian function of the above optimization problem as

$$
\begin{aligned}
L(f,\mu,\gamma) &= \frac{1}{2n\eta}\sum_{i=1}^{n}\xi_i + \frac{1}{2}\sum_{j=1}^{d}\tau_j\langle f^{(j)}, f^{(j)}\rangle_{K^{(j)}} - \sum_{i=1}^{n}\gamma_i\xi_i \\
&\quad - \sum_{i=1}^{n}\mu_i\Big(y_i\sum_{j=1}^{d}\langle f^{(j)}, K^{(j)}(x_i^{(j)},\cdot)\rangle_{K^{(j)}} - 1 + \xi_i\Big),
\end{aligned}
$$

where $\mu_i, \gamma_i, 1 \le i \le n$ are Lagrangian parameters.

The parameters minimizing $L$ satisfy

$$\frac{\partial L}{\partial f^{(j)}} = \tau_j f^{(j)} - \sum_{i=1}^{n} \mu_i y_i K^{(j)}(x_i^{(j)}, \cdot) = 0, \forall j \in \{1, ..., d\},$$

$$\frac{\partial L}{\partial \xi_i} = \frac{1}{2n\eta} - \beta_i - \gamma_i = 0.$$

Then, we obtain that

$$f^{(j)} = \sum_{i=1}^{n} \mu_i \tau_j^{-1} y_i K^{(j)}(x_i^{(j)}, \cdot), \ \ \forall j \in \{1, ..., d\},$$

and

$$\mu_i + \gamma_i = \frac{1}{2n\eta}, \ \ \forall i \in \{1, ..., n\}.$$

Hence, $\bar{f}_{\mathbf{z}}$ defined in Section 2 satisfies that

$$\bar{f}_{\mathbf{z}} = \sum_{j=1}^{d} \bar{f}_{\mathbf{z}}^{(j)} = \sum_{j=1}^{d}\sum_{i=1}^{n} \mu_i \tau_j^{-1} y_i K^{(j)}(x_i^{(j)}, \cdot) \tag{4}$$

with

$$0 \le \mu_i \le \frac{1}{2n\eta}, \forall i \in \{1, ..., n\}. \tag{5}$$

**Proposition 2** *For the hypothesis error $E_3$ defined in (3), there holds*

$$E_3 \le \frac{\lambda d}{2\eta\sqrt{n}}.$$

**Proof.** From the definitions of $f_{\mathbf{z}}$ and $\bar{f}_{\mathbf{z}}$ in Section 2, we know that

$$\mathcal{E}_{\mathbf{z}}(f_{\mathbf{z}}) \le \mathcal{E}_{\mathbf{z}}(f_{\mathbf{z}}) + \lambda\Omega(f_{\mathbf{z}}) \le \mathcal{E}_{\mathbf{z}}(\bar{f}_{\mathbf{z}}) + \lambda\Omega(\bar{f}_{\mathbf{z}})$$

and

$$\mathcal{E}_{\mathbf{z}}(\bar{f}_{\mathbf{z}}) + \eta\sum_{j=1}^{d}\tau_j\|\bar{f}_{\mathbf{z}}^{(j)}\|_{K^{(j)}}^2 \le \mathcal{E}_{\mathbf{z}}(f_\eta) + \eta\sum_{j=1}^{d}\tau_j\|f_\eta^{(j)}\|_{K^{(j)}}^2.$$

Then,

$$
\begin{aligned}
E_3 &= \mathcal{E}_{\mathbf{z}}(f_{\mathbf{z}}) + \lambda\Omega(f_{\mathbf{z}}) - \left(\mathcal{E}_{\mathbf{z}}(f_\eta) + \eta\sum_{j=1}^{d}\tau_j\|f_\eta^{(j)}\|_{K^{(j)}}^2\right) \\
&\le \mathcal{E}_{\mathbf{z}}(\bar{f}_{\mathbf{z}}) + \lambda\Omega(\bar{f}_{\mathbf{z}}) - \left(\mathcal{E}_{\mathbf{z}}(f_\eta) + \eta\sum_{j=1}^{d}\tau_j\|f_\eta^{(j)}\|_{K^{(j)}}^2\right) \\
&\le \mathcal{E}_{\mathbf{z}}(\bar{f}_{\mathbf{z}}) + \eta\sum_{j=1}^{d}\tau_j\|\bar{f}_{\mathbf{z}}^{(j)}\|_{K^{(j)}}^2 - \left(\mathcal{E}_{\mathbf{z}}(f_\eta) + \eta\sum_{j=1}^{d}\tau_j\|f_\eta^{(j)}\|_{K^{(j)}}^2\right) + \lambda\Omega(\bar{f}_{\mathbf{z}}) \\
&\le \lambda\Omega(\bar{f}_{\mathbf{z}}).
\end{aligned}
\tag{6}
$$

According to $\bar{f}_{\mathbf{z}}$ in (4) and (5), we have

$$\lambda\Omega(\bar{f}_{\mathbf{z}}) = \lambda\sum_{j=1}^{d}\tau_j\sqrt{\sum_{i=1}^{n}(\mu_i\tau_j^{-1})^2} = \lambda\sum_{j=1}^{d}\tau_j\sqrt{\sum_{i=1}^{n}\mu_i^2} \le \frac{\lambda d}{2\eta\sqrt{n}}. \tag{7}$$

Combining (6) and (7), we derive the desired estimate. $\square$

## 1.2 Sample error estimate

The error term $E_1$ in (1) reflects the divergence between the expected excess risk $\mathcal{E}(\pi(f_{\mathbf{z}})) - \mathcal{E}(f_c)$ and the empirical excess risk $\mathcal{E}_{\mathbf{z}}(\pi(f_{\mathbf{z}})) - \mathcal{E}_{\mathbf{z}}(f_c)$. Since $f_{\mathbf{z}}$ involves any given $\mathbf{z} = \{(x_i, y_i)\}_{i=1}^n$, we introduce the concentration inequality in [5] to bound $E_1$.

**Lemma 1** *Let $\mathcal{G}$ be a set of measurable functions on $\mathcal{Z}$ and $B, c > 0$, $\tau \in [0,1]$ be constants such that $\|g\|_\infty \leq B$, $Eg^2 \leq c(Eg)^\tau$ for all $g \in \mathcal{G}$. Assume that $\log \mathcal{N}_2(\mathcal{G}, \varepsilon) \leq a\varepsilon^{-s}, \forall \varepsilon > 0$ for some $a > 0$ and $s \in (0,2)$. Then, there exists a constant $c_s'$ such that for any $\delta \in (0,1)$*

$$Eg - \frac{1}{n}\sum_{i=1}^n g(z_i) \leq \frac{1}{2}\zeta^{1-\tau}(Eg)^\tau + c_s'\zeta + 2\left(\frac{c\log(1/\delta)}{n}\right)^{\frac{1}{2-\tau}} + \frac{18B\log(1/\delta)}{n}$$

*with confidence $1 - \delta$, where*

$$\zeta = \max\left\{ c^{\frac{2-s}{4-2\tau+s\tau}}\left(\frac{a}{n}\right)^{\frac{2}{4-2\tau+s\tau}}, B^{\frac{2-s}{2+s}}\left(\frac{a}{n}\right)^{\frac{2}{2+s}} \right\}.$$

The following lemma demonstrates the upper bound of $f_{\mathbf{z}}$ for any $\mathbf{z} \in \mathcal{Z}^n$.

**Lemma 2** *For $f_{\mathbf{z}}$ defined in Section 2, there holds*

$$\|f_{\mathbf{z}}^{(j)}\|_{K^{(j)}} \leq \|f_{\mathbf{z}}\|_K \leq \frac{\kappa\sqrt{n}}{\lambda \min_j \tau_j}, \forall j \in \{1, ..., d\}.$$

**proof.** The definition $f_{\mathbf{z}}$ tells us that

$$\Omega(f_{\mathbf{z}}) = \sum_{j=1}^d \tau_j \sqrt{\sum_{i=1}^n (\alpha_{\mathbf{z},i}^{(j)})^2} \leq \frac{1}{\lambda}.$$

This means

$$\sum_{j=1}^d \sqrt{\sum_{i=1}^n (\alpha_{\mathbf{z},i}^{(j)})^2} \leq \frac{1}{\lambda \min \tau_j}. \tag{8}$$

Meanwhile, we deduce that

$$\|f_{\mathbf{z}}\|_K \leq \sum_{j=1}^d \|f_{\mathbf{z}}^{(j)}\|_{K^{(j)}} \leq \kappa \sum_{j=1}^d \sum_{i=1}^n |\alpha_{\mathbf{z},i}^{(j)}| \leq \kappa\sqrt{n}\sum_{j=1}^d \sqrt{\sum_{i=1}^n (\alpha_{\mathbf{z},i}^{(j)})^2}, \tag{9}$$

where the last inequality follows from Höder inequality.

The desired upper bound follows by combining (8) and (9). $\square$

**Proposition 3** *Under Assumptions A and B, for any $\delta \in (0,1)$, there holds*

$$
\begin{aligned}
E_1 &\leq C_1 \max\left\{ \lambda^{\frac{-2s(q+1)}{4+2q+sq}} n^{-\frac{(2-s)(q+1)}{4+2q+sq}}, \lambda^{-\frac{2s}{2+s}} n^{-\frac{2-s}{2+s}} \right\}^{\frac{1}{1+q}} \left(\mathcal{E}(\pi(f_{\mathbf{z}})) - \mathcal{E}(f_c)\right)^{\frac{q}{1+q}} \\
&\quad + C_2 \max\left\{ \lambda^{\frac{-2s(q+1)}{4+2q+sq}} n^{-\frac{(2-s)(q+1)}{4+2q+sq}}, \lambda^{-\frac{2s}{2+s}} n^{-\frac{2-s}{2+s}} \right\} + \frac{36\log(1/\delta)}{n} \\
&\quad + \left(8(2\Delta)^{-\frac{q}{q+1}}\log(1/\delta)\right)^{\frac{q+1}{q+2}} n^{-\frac{q+1}{q+2}}
\end{aligned}
$$

*with confidence $1 - \delta$, where $C_1, C_2$ are positive constants independent of $n$.*

**Proof.** Recall that

$$E_1 = \int [(1 - y\pi(f_{\mathbf{z}})(x))_+ - (1 - yf_c(x))_+]d\rho - \frac{1}{n}\sum_{i=1}^n [(1 - y_i\pi(f_{\mathbf{z}})(x_i))_+ - (1 - y_if_c(x_i))_+]$$

and $f_{\mathbf{z}} \in \mathcal{B}_r$ with $r = \frac{\kappa\sqrt{n}}{\lambda \min_j \tau_j}$. We consider the function set

$$\mathcal{G} = \left\{ g(z) = (1 - y\pi(f)(x))_+ - (1 - yf_c(x))_+ : f \in \mathcal{B}_r, (x, y) \in \mathcal{Z} \right\}.$$

Since for any $f_1, f_2 \in \mathcal{B}_r$

$$|(1 - y\pi(f_1)(x))_+ - (1 - y\pi(f_2)(x))_+| \leq |y\pi(f_1)(x) - y\pi(f_2)(x)| \leq |f_1(x) - f_2(x)|,$$

we have

$$\log \mathcal{N}_2(\mathcal{G}, \varepsilon) \leq \log \mathcal{N}_2(\mathcal{B}_r, \varepsilon) \leq \log \mathcal{N}_2(\mathcal{B}_1, \varepsilon r^{-1}) \leq c_s d^{1+s} r^s \varepsilon^{-s},$$

where the last inequality follows from Assumption B.

Considering $0 \leq (1 - y\pi(f)(x))_+ \leq 2$ and $0 \leq (1 - yf_c(x))_+ \leq 2$, we get that $\|g\|_\infty \leq 2$ for every $g \in \mathcal{G}$. Under Assumption A, there holds

$$Eg^2 \leq 8(2\Delta)^{-\frac{q}{q+1}}(Eg)^{\frac{q}{1+q}}.$$

Hence, we can apply Lemma 1 to get the concentration estimate for any $g \in \mathcal{G}$, where the parameters $a = c_s d^{1+s} r^s$, $B = 2$, $c = 8(2\Delta)^{-\frac{q}{q+1}}$, and $\tau = \frac{q}{q+1}$. Note that $f_{\mathbf{z}} \in \mathcal{B}_r$ with $r = \frac{\kappa\sqrt{n}}{\lambda \min_j \tau_j}$. We deduce that

$$E_1 \leq \frac{1}{2}\zeta^{\frac{1}{q+1}}\left(\mathcal{E}(\pi(f_{\mathbf{z}})) - \mathcal{E}(f_c)\right)^{\frac{q}{1+q}} + \frac{36\log(1/\delta)}{n} + c_s'\zeta + \left(\frac{8(2\Delta)^{-\frac{q}{q+1}}\log(1/\delta)}{n}\right)^{\frac{q+1}{q+2}},$$

with confidence $1 - \delta$, where

$$\zeta = \max\left\{ 2^{\frac{(2-s)(1+q)}{4+2q+sq}}(c_s\kappa^s d^{s+s^2}(\min\tau_j)^{-s})^{\frac{2q+2}{4+2q+sq}}\lambda^{\frac{-2s(q+1)}{4+2q+sq}} n^{-\frac{(2-s)(q+1)}{4+2q+sq}}, \right.$$

$$\left. 2^{\frac{2-s}{2+s}} c_s^{\frac{2}{2+s}} d^{\frac{2+2s}{2+s}}(\kappa)^{\frac{2s}{2+s}}(\lambda\min\tau_j)^{-\frac{2s}{2+s}} n^{-\frac{2-s}{2+s}} \right\}.$$

This completes the proof. $\square$

Now we turn to bound the error term $E_2$ in terms of the following one-side Bernstein inequality [1, 3, 4].

**Lemma 3** *Let $\xi$ be a random variable on a probability space $\mathcal{Z}$ satisfying $|\xi(z) - E\xi| \leq M_\xi$ for some constant $M_\xi$ and $\sigma$ is its variance. Then, for any $\delta \in (0, 1)$, with confidence $1 - \delta$ there holds*

$$\frac{1}{n}\sum_{i=1}^n \xi(z_i) - E\xi \leq \frac{2M_\xi \log(1/\delta)}{3n} + \sqrt{\frac{2\sigma^2\log(1/\delta)}{n}}.$$

**Proposition 4** *Under Assumption A, for any $\delta \in (0, 1)$, we have*

$$E_2 \leq \frac{5\kappa\log(1/\delta)}{3n}\sqrt{\frac{D(\eta)}{\eta\min_j \tau_j}} + \frac{4\log(1/\delta)}{3n} + \frac{q+2}{2q+2}\left(\frac{16(2\Delta)^{-\frac{q}{q+1}}\log(1/\delta)}{n}\right)^{\frac{q+1}{q+2}} + D(\eta)$$

*with confidence $1 - 2\delta$.*

**Proof.** To bound $E_2$, we introduce

$$\xi(z) = (1 - yf_\eta(x))_+ - (1 - yf_c(x))_+, z = (x, y) \in \mathcal{Z}.$$

It is easy to verify that

$$E_2 = \frac{1}{n}\sum_{i=1}^n \xi(z_i) - E\xi = \left\{\frac{1}{n}\sum_{i=1}^n \xi_1(z_i) - E\xi_1\right\} + \left\{\frac{1}{n}\sum_{i=1}^n \xi_2(z_i) - E\xi_2\right\}, \tag{10}$$

where

$$\xi_1(z) = (1 - yf_\eta(x))_+ - (1 - y\pi(f_\eta)(x))_+$$

and

$$\xi_2 = (1 - y\pi(f_\eta)(x))_+ - (1 - yf_c(x))_+.$$

The definition $f_\eta$ in Section 3 tells us that

$$\|f_\eta\|_\infty \leq \kappa \|f_\eta\|_K \leq \kappa \sqrt{\frac{D(\eta)}{\eta \min_j \tau_j}}.$$

Then, for any $z \in \mathcal{Z}$,

$$0 \leq \xi_1(z) \leq |f_\eta(x) - \pi(f_\eta)(x)| \leq \kappa \sqrt{\frac{D(\eta)}{\eta \min_j \tau_j}},$$

$|\xi_1 - E\xi_1| \leq \kappa \sqrt{\frac{D(\eta)}{\eta \min_j \tau_j}}$, and $\sigma^2(\xi_1) \leq \kappa \sqrt{\frac{D(\eta)}{\eta \min_j \tau_j}} E\xi_1$.

Applying Lemma 3 to $\xi_1$, we obtain with confidence $1 - \delta$

$$
\begin{aligned}
\frac{1}{n}\sum_{i=1}^n \xi_1(z_i) - E\xi_1 &\leq \frac{2\kappa \log(1/\delta)}{3n}\sqrt{\frac{D(\eta)}{\eta \min_j \tau_j}} + \sqrt{\frac{2\kappa E\xi_1 \log(1/\delta)}{n}}\sqrt{\frac{D(\eta)}{\eta \min_j \tau_j}} \\
&\leq \frac{5\kappa \log(1/\delta)}{3n}\sqrt{\frac{D(\eta)}{\eta \min_j \tau_j}} + E\xi_1. \qquad (11)
\end{aligned}
$$

Now we turn to consider the concentration estimate of $\xi_2$. Note that $0 \leq \xi_2(z) \leq 2$ for any $z \in \mathcal{Z}$. Applying Lemma 3 to $\xi_2$, we get with confidence $1 - \delta$

$$
\begin{aligned}
\frac{1}{n}\sum_{i=1}^n \xi_2(z_i) - E\xi_2 &\leq \frac{4 \log(1/\delta)}{3n} + \sqrt{\frac{2 E\xi_2^2 \log(1/\delta)}{n}} \\
&\leq \frac{4 \log(1/\delta)}{3n} + \sqrt{\frac{16(2\Delta)^{-\frac{q}{q+1}} \log(1/\delta)}{n}} (E\xi_2)^{\frac{q}{2q+2}}, \qquad (12)
\end{aligned}
$$

where the last inequality follows form Assumption A.

Recall that

$$\frac{1}{t} + \frac{1}{t'} = 1 \text{ with } t, t' > 0 \Rightarrow a \cdot b \leq \frac{a^t}{t} + \frac{b^{t'}}{t'}, \forall a, b \geq 0.$$

Applying this elementary inequality to $a = \sqrt{\frac{16(2\Delta)^{-\frac{q}{q+1}} \log(1/\delta)}{n}}, b = (E\xi_2)^{\frac{q}{2q+2}}, t = \frac{2q+2}{q+2}, t' = \frac{2q+2}{q}$, we further get

$$\sqrt{\frac{16(2\Delta)^{-\frac{q}{q+1}} \log(1/\delta)}{n}} (E\xi_2)^{\frac{q}{2q+2}} \leq \frac{q+2}{2q+2}\left(\frac{16(2\Delta)^{-\frac{q}{q+1}} \log(1/\delta)}{n}\right)^{\frac{q+1}{q+2}} + \frac{q}{2q+2} E\xi_2.$$

This together with (12) means

$$\frac{1}{n}\sum_{i=1}^n \xi_2(z_i) - E\xi_2 \leq \frac{4 \log(1/\delta)}{3n} + \frac{q+2}{2q+2}\left(\frac{16(2\Delta)^{-\frac{q}{q+1}} \log(1/\delta)}{n}\right)^{\frac{q+1}{q+2}} + \frac{q}{2q+2} E\xi_2. \quad (13)$$

Combining (10), (11) and (13), we obtain, with confidence $1 - 2\delta$,

$$E_2 \leq \frac{5\kappa \log(1/\delta)}{3n}\sqrt{\frac{D(\eta)}{\eta \min_j \tau_j}} + \frac{4 \log(1/\delta)}{3n} + \frac{q+2}{2q+2}\left(\frac{16(2\Delta)^{-\frac{q}{q+1}} \log(1/\delta)}{n}\right)^{\frac{q+1}{q+2}} + E\xi.$$

The desired result follows by considering $E\xi \leq D(\eta)$. $\quad \square$

### 1.3 Proof of Theorem 1

**Proof.** Combining Propositions 1-4, we get with confidence $1 - 3\delta$

$$
\begin{aligned}
&\mathcal{E}(\pi(f_{\mathbf{z}})) - \mathcal{E}(f_c) \\
\leq \quad &C_1 \max\left\{\lambda^{\frac{-2s(q+1)}{4+2q+sq}} n^{-\frac{(2-s)(q+1)}{4+2q+sq}}, \lambda^{-\frac{2s}{2+s}} n^{-\frac{2-s}{2+s}}\right\}^{\frac{1}{1+q}} \left(\mathcal{E}(\pi(f_{\mathbf{z}})) - \mathcal{E}(f_c)\right)^{\frac{q}{1+q}} \\
&+C_2 \max\left\{\lambda^{\frac{-2s(q+1)}{4+2q+sq}} n^{-\frac{(2-s)(q+1)}{4+2q+sq}}, \lambda^{-\frac{2s}{2+s}} n^{-\frac{2-s}{2+s}}\right\} + \left(\frac{24(2\Delta)^{-\frac{q}{q+1}} \log(1/\delta)}{n}\right)^{\frac{q+1}{q+2}} \\
&+\frac{112\log(1/\delta)}{n} + \frac{5\kappa\log(1/\delta)}{3n}\sqrt{\frac{D(\eta)}{\eta\min_j \tau_j}} + \frac{q+2}{2q+2}\left(\frac{16(2\Delta)^{-\frac{q}{q+1}}\log(1/\delta)}{n}\right)^{\frac{q+1}{q+2}} \\
&+2D(\eta) + \frac{d\lambda}{2\eta\sqrt{n}}.
\end{aligned}
$$

Recall that, for $a, b > 0$ and $t \in (0,1)$,

$$
x \leq ax^t + b, x > 0 \Rightarrow x \leq \max\{(2a)^{\frac{1}{1-t}}, 2b\}.
$$

We apply the above elementary inequality to $x = \mathcal{E}(\pi(f_{\mathbf{z}})) - \mathcal{E}(f_c)$ and $t = \frac{q}{q+1}$. Then, under Assumption C, we further get

$$
\begin{aligned}
\mathcal{E}(\pi(f_{\mathbf{z}})) - \mathcal{E}(f_c) \quad \leq \quad &C\log(3/\delta)\Big( \max\left\{\lambda^{\frac{-2s(q+1)}{4+2q+sq}} n^{-\frac{(2-s)(q+1)}{4+2q+sq}}, \lambda^{-\frac{2s}{2+s}} n^{-\frac{2-s}{2+s}}\right\} \\
&+n^{-\frac{q+1}{q+2}} + \eta^{\frac{\beta-1}{2}} n^{-1} + \eta^{\beta} + \lambda\eta^{-1} n^{-\frac{1}{2}} \Big)
\end{aligned}
\tag{14}
$$

with confidence $1 - \delta$.

Setting $\eta^{\beta} = \lambda\eta^{-1} n^{-\frac{1}{2}}$, we have $\eta = \lambda^{\frac{1}{\beta+1}} n^{-\frac{1}{2\beta+2}}$. Then, (14) yields with confidence $1 - \delta$

$$
\begin{aligned}
&\mathcal{E}(\pi(f_{\mathbf{z}})) - \mathcal{E}(f_c) \\
\leq \quad &C\log(3/\delta)\left(\lambda^{\frac{-2s(q+1)}{4+2q+sq}} n^{-\frac{(2-s)(q+1)}{4+2q+sq}} + n^{-\frac{q+1}{q+2}} + \lambda^{\frac{\beta-1}{2\beta+2}} n^{-\frac{3+5\beta}{4+4\beta}} + \lambda^{\frac{\beta}{\beta+1}} n^{-\frac{\beta}{2\beta+2}}\right),
\end{aligned}
\tag{15}
$$

where $C$ is a positive constant independent of $n, \delta$.

The desired result follows by taking $\lambda = n^{-\theta}$ with $\theta \in (0, \min\{\frac{2-s}{2s}, \frac{3+5\beta}{2-2\beta}\})$ in (15) and considering

$$
\mathcal{R}(\text{sgn}(f_{\mathbf{z}})) - \mathcal{R}(f_c) \leq \mathcal{E}(\pi(f_{\mathbf{z}})) - \mathcal{E}(f_c).
$$