[Reviews · NeurIPS 2017]

Reviewer 1



The aim of the paper is to discuss and analyze the use of learning model with grouped variables, which seems to be very useful in feature classification problems but has never be considered in the context of the RKHS. This paper gives a contribution which is mainly theoretical, it finds a bound to the generalization error and demonstrates that, under mild conditions, the learning rate can be polynomial Theorem and Proofs the mathematical part/assumptions are clear and well explained. The proof lies on the point (A), (B) and (C), the first one is a kind of convergence in probability related to noise, the others are related to the space of functions necessary for recovering with good approximation the true function. The theoretical part ends with the Theorem 1, which shows that the method can reach a polynomial convergence rate to the true solution Consequences from the paper seems that no one else has studied the asymptotic theory of sparse group problem for the classification task, neither the learning rate of this algorithm. This part ends with a sketch of the proof Experiments the authors show the result of GroupSAM on an artificial data, built in a relatively low dimensional case. From this point of view the results are not super self explainatory of what affirmed in the abstract variable selection capability for real world applications which focus on high-dimensional space (400 samples vs 24 dimensions). The authors cite another paper which followed the same path in the generation of the data set and the division if the groups (not clear to me the replication of the features at line (243)

Reviewer 2



This paper presents a new additive classification model (GroupSAM) with grouped variables based on group based additive models in kernel Hilbert spaces, and an upper bound for the generalization error is provided in Theorem 1 to show that GroupSAM converges to the optimal error with a polynomial decay under certain conditions. Experiments are performed on synthetic data and seven UCI datasets that have been modified in order to get some group structure in the feature space. The results provided in the paper show that the proposed method is able to take into account grouped features while yielding good performance in terms of classification accuracy. As far as I know, the paper presents a novel method for group sparse nonlinear classification with additive models. However, the degree of contribution of this paper with respect to Yin, Junming, Xi Chen, and Eric P. Xing. "Group sparse additive models." Proceedings of the... International Conference on Machine Learning. International Conference on Machine Learning. Vol. 2012. NIH Public Access, 2012. Is not clear to me as the framework presented in the above paper seems to be easy to extend to classification settings. In fact, Yin et al. apply their group sparse additive model for regression to address a classification task (classification of breast cancer data) by taking the sign of the predicted responses. On the other hand, I appreciate authors’ efforts in providing a theoretical analysis on the generalization error bound of GroupSAM and tis convergence. Regarding the empirical results, I do not think that the experiments show the real significance of the proposed method as they real-world data has been “arbitrarily” tuned to enforce group structure. I think that the use of real data with group structure by itself is mandatory in order to properly measure the contribution of this work. For example, in bioinformatics in can be useful to build predictive models that take into account genes’ pathway information. I also miss comparisons with other parametric approaches such as Group Lasso or the GroupSpSAM model adapted to the classification setting. Additionally, I cannot see a clear advantage with respect to Gaussian SVMs in terms of classification accuracy. I also would like to know whether the Gaussian kernel parameter was properly tuned. Finally, I would like to know how multiclass classification was performed in the toy example as the GroupSAM method is formulated in Section 2 for binary classification problems.  Overall, the paper is well written and it has a well structure. However, I think that the introduction does not clearly present the differences between the proposed approach and other sparse additive models and group sparse models in parametric settings (e.g. Group Lasso).

Reviewer 3



This paper studied proposed group sparse additive machines in the classification context and studied their generalization properties. The group sparse additive machine (GroupSAM) was proposed by exploring and utilizing the structural information among the input variables. The newly proposed classification model works in a data-dependent hypothesis space and so possesses some flexibility properties. Explicit generalization bounds were also established. The paper is well written. The presented analysis seems to be correct and the experimental results looks convincing.

Reviewer 4



(I'm a back-up reviewer called in last minute. I'm knowledgeable in this field but not an expert.) SUMMARY: The paper studied a group SVM model where the coordinates are divided into groups (known a priori), and each group is equipped with a separate kernel space. Finally, the regularizer is the sum of the l2 norms (no squares) of the predictor of each group, and this ensures sparsity in the number of groups. NOVELTY: The "group" idea and the "sum of l2 norm" idea are not new (see [24, ICML 2012]); however, the study of the generalization bounds with the presence of (1) kernels, and (2) the hinge loss (as opposed to squared loss) is new. I didn't verify the proofs in the appendix (for the generalization bound, Theorem 1). It is non-trivial, looks convincing, and the techniques follow from citation [5] in the appendix. The main "delta" between the two proofs seem to be how to deal with the sum of l2 norms, that is Lemma 2 in the appendix. IN SUM: I view the generalization bound as "a bound" but not necessarily the best bound, because the authors did not provide evidence why it should be tight. This paper deserves publication. However, the lack of novelty in the model and the proof makes me hard to vote for a strong accept.